# Capsule Endoscopy in Inflammatory Bowel Disease: When? To Whom?

**DOI:** 10.3390/diagnostics11122240

**Published:** 2021-11-30

**Authors:** Soo-Young Na, Yun-Jeong Lim

**Affiliations:** 1Department of Internal Medicine, Incheon St. Mary’s Hospital, College of Medicine, The Catholic University of Korea, Incheon 21431, Korea; sktndud@hanmail.net; 2Department of Internal Medicine, Dongguk University Ilsan Hospital, Dongguk University College of Medicine, Goyang 10326, Korea

**Keywords:** inflammatory bowel disease, Crohn’s disease, colitis, ulcerative, capsule endoscopy, colon capsule endoscopy

## Abstract

Capsule endoscopy (CE) has proven to be a valuable diagnostic modality for small bowel diseases over the past 20 years, particularly Crohn’s disease (CD), which can affect the entire gastrointestinal tract from the mouth to the anus. CE is not only used for the diagnosis of patients with suspected small bowel CD, but can also be used to assess disease activity, treat-to-target, and postoperative recurrence in patients with established small bowel CD. As CE can detect even mildly non-specific small bowel lesions, a high diagnostic yield is not necessarily indicative of high diagnostic accuracy. Moreover, the cost effectiveness of CE as a third diagnostic test employed usually after ileocolonoscopy and MR or CT enterography is an important consideration. Recently, new developments in colon capsule endoscopy (CCE) have increased the utility of CE in patients with ulcerative colitis (UC) and pan-enteric CD. Although deflation of the colon during the examination and the inability to evaluate dysplasia-associated lesion or mass results in an inherent risk of overestimation or underestimation, the convenience of CCE examination and the risk of flare-up after colonoscopy suggest that CCE could be used more actively in patients with UC.

## 1. Introduction

Capsule endoscopy (CE), which was first used to evaluate the small intestine in 10 healthy volunteers by Iddan et al. [1], has enabled direct visual observation of small bowel lesions and phenotypes that had been difficult to assess using conventional endoscopy [2]. As CE allows direct observation of the small intestine, it is able to visualize even mildly inflammatory mucosal lesions, such as erythema, erosion, and small ulcers, which are difficult to detect with radiological imaging modalities such as small bowel follow-through (SBFT), small bowel contrast ultrasound (SBCUS), CT enterography (CTE), and MR enterography (MRE) [3]. This advantage has aided in precision medicine-based diagnostic and therapeutic decision-making, especially in patients with suspected or established Crohn’s disease (CD) of the small intestine. The scope of use of CE has expanded over the past 20 years, allowing its application in patients with ulcerative colitis (UC), along with pan-enteric CD. This has been made possible with the subsequent development of colon capsule endoscopy (CCE), which enables visualization of both the small and large intestines [4].

The incidence of inflammatory bowel disease (IBD) is relatively high but rather stable in Western countries; however, the incidence and prevalence of IBD are rapidly increasing in Asia, Eastern Europe, and South America, and this shift in the epidemiology of IBD indicates that it has become a global disease, whose increasing incidence has been burdening health services in recent years [5,6,7,8]. Therefore, the use of CE for the diagnosis and management of IBD is becoming more frequent and its implementation is considered a priority in the field of IBD. This review will describe the clinical applications and value of CE in the diagnosis and management of inflammatory bowel disease.

## 2. Crohn’s Disease

Patients with suspected CD (SCD) who cannot be diagnosed by radiological modalities can be diagnosed by small bowel CE (SBCE), since the latter can visualize even mildly superficial mucosal lesions that are rarely visible using radiological imaging techniques [9]. SBCE is also useful in assessing disease activity, treat-to-target, and postoperative recurrence in patients with established CD (ECD).

CD mainly involves terminal ileum, and the disease location is limited to the small bowel alone in about 30% of patients [10,11]. Ileocolonoscopy with biopsy is considered the first-line diagnostic modality in patients with suspected CD. However, lesions located proximal to the terminal ileum are difficult to diagnose by conventional ileocolonoscopy. Patients with small bowel CD (SBCD) have been diagnosed radiologically by SBFT or small bowel enteroclysis, which are modalities with reasonable diagnostic accuracy [12,13]. Recently, however, these modalities have been largely replaced by cross-sectional imaging modalities, which can better classify disease phenotype and behavior. Several meta-analyses have shown that CTE, MRE, and SBCUS have similar accuracy in the diagnosis of SBCD [14]. SBCE, a safe and painless endoscopic method for the evaluation of the small bowel, is useful diagnostic modality for SBCD. The results of a Spanish national survey showed that SBCE is widely used to manage IBD in patients with SCD (76.3%), to assess inflammatory activity (54.7%) and to evaluate the extent of disease (54.7%) [15]. In addition, CE has been shown to be useful in assessing mucosal CD activity in selected patients with colonic CD [16]. The typical CE findings in SBCD are shown in Figure 1A.

### 2.1. Capsule Retention

Although CE is a simple and safe test, capsule retention (CR) in the gastrointestinal tract is a frequent complication. CE can be retained within the small bowel, usually in patients with stricturing disease. Passage of the CE through the gastrointestinal tract, including the pylorus and ileocecal valve, may be difficult in pediatric patients, but this method can be safely used in children over 9 years of age [18].

A meta-analysis that included 25 studies of 5876 patients with obscure gastrointestinal bleeding, nine studies of 968 patients with SCD, and 11 studies of 558 patients with ECD found that the pooled CR rates were 3.6% (95% confidence interval (CI), 1.7–8.6%) for SCD and 8.2% (95% CI, 6.0–11.0%) for ECD [19]. These CR rates decreased to 2.7% (95% CI, 1.1–6.4%) in subsequent CE after a patency capsule (PC) or CTE to exclude strictures, with the latter comparable to the CR rate for obscure gastrointestinal bleeding (2.1%; 95% CI, 1.5–2.8%) [19]. A more recent meta-analysis found that the CR rates were 2.35% (95% CI, 1.31–4.19%) in 1234 patients with SCD and 4.63% (95% CI, 3.42–6.25%) in 1720 patients with ECD [20]. Although these CR rates are significantly lower than those previously reported, this meta-analysis also found that the CR rate after PC in patients with ECD was 2.88% (95% CI, 1.74–4.74%) [20].

A meta-analysis that included five studies of 203 patients showed that PC had a sensitivity of 97% (95% CI, 93–99%), a specificity of 83% (95% CI, 65–95%), and an accuracy of 0.956 in diagnosing small bowel obstruction [21]. A recent meta-analysis found that the CR rates in patients with ECD were 2.32% (95% CI, 0.87–6.03%) after negative small bowel cross-sectional imaging and 2.88% (95% CI, 1.74–4.74%) after negative PC [20]. MRE showed a high sensitivity (>92%) and negative predictive value (NPV) (>96%) for PC retention in patients with ECD [22]. These results suggest that cross-sectional imaging of the small bowel should be performed prior to CE to determine the presence of strictures in patients with CD. However, a multicenter prospective study showed that CR rates are similar in low-risk (0.7%, 20/2942) and negative PC high-risk (0.7%, 1/151) patients, but are significantly higher in high-risk patients with negative cross-sectional imaging (CTE or MRE; 8.3%, 2/24; *p* = 0.049) [23]. Thus, even if cross-sectional imaging is normal, the PC procedure should be offered to patients at increased risk of CR.

As CR is usually asymptomatic, initial periodic monitoring is suggested. Even if CR has occurred in IBD patients, the European Society of Gastrointestinal Endoscopy guidelines recommend observation in patients with asymptomatic CR for the following reasons [24]: (1) 35–50% of CR patients spontaneously excreted the capsules after ≥15 days without further management, (2) a short course of medical therapy may allow capsule excretion, and (3) spontaneous excretion usually occurs 4–12 weeks after ingestion [25]. However, device-assisted enteroscopy or surgical removal should be considered if the capsule is not excreted after 3–6 months or if patients experience symptoms of acute obstruction [25].

### 2.2. Superior Diagnostic Yield to Other Imaging Modalities

CE provides high-resolution endoluminal images of the small bowel. As there is no standard modality that can be compared with CE to determine its diagnostic accuracy, the ‘diagnostic yield’ is determined in many studies of this modality.

CE showed a higher diagnostic yield than small bowel enteroclysis in the detection of small bowel lesions, in 27 patients with ECD (74.1% vs. 40.7%; *p* < 0.05) and in 20 patients with SCD (65% vs. 30%; *p* < 0.05) [26]. CE also showed a higher diagnostic yield than CT enterolysis in evaluating jejunal or ileal lesions in 41 patients with small bowel CD (61.0% vs. 29.3%; *p* < 0.004) but not in those with terminal/neonatal ileum [27]. CE was superior to MR enteroclysis in detecting inflammatory lesions in the proximal and middle parts of the small bowel (66.7% vs. 5.6%; *p* = 0.016) in 18 patients with SCD or ESD [28].

A meta-analysis that included only prospective studies found that the diagnostic yield of CE was significantly superior to that of SBFT, CTE, ileocolonoscopy, and push enteroscopy in the evaluation of SBCD [29]. The results of recent meta-analysis have revealed that the diagnostic yields of CE do not differ significantly from those of CTE and MRE in patients with both SCD and ECD [30]. The different results for CTE in these meta-analyses were due to differences in the inclusion criteria. Another recent meta-analysis found that the diagnostic yield of CE was similar to that of MRE in 10 studies involving 400 patients (odds ratio (OR), 1.17; 95% CI, 0.83–1.67) and SBCUS in five studies involving 142 patients (OR, 0.88; 95% CI, 0.51–1.53) with SCD and ECD [31]. In that analysis, however, CE was superior to MRE in seven studies involving 251 patients with proximal suspected small bowel disease (OR, 2.79; 95% CI, 1.2–6.48) [31]. Subsequent prospective studies comparing CE with MRE found that both modalities independently detected previously unrecognized proximal disease locations in 51% (29/56) and 26% (20/79) (*p* < 0.01), respectively, in patients with SBCD in clinical remission or with mild disease [32]. CE was also superior to MRE in detecting small bowel lesions (76.6% vs. 44.7%, *p* = 0.001) in SCD and ECD, with CE being superior to MRE in detecting lesions in the jejunum, ileum, and terminal ileum (all, *p* < 0.05) [9].

Although the diagnostic yield of CE was higher than that of other imaging diagnostic modalities, especially in patients with proximal small bowel disease, there are several barriers to the clinical use of CE. The higher diagnostic yield of CE does not directly indicate a higher diagnostic accuracy, as diagnostic yield can be affected by other factors, such as nonsteroidal anti-inflammatory drugs [33]. Minor mucosal injuries and erosions have been detected in up to 20% of healthy volunteers [33]. In addition, cost analyses suggest that the addition of CE as a third test after ileocolonoscopy and negative CTE or SBFT is not cost-effective [34].

### 2.3. Increasing Diagnostic Capability of Capsule Endoscopy

CE is being developed in several ways, including the development of CE instruments with higher frame rates and increased image resolution, which should increase the possibility of obtaining higher diagnostic yield and accuracy than in the past [35]. For example, the adaptive frame rate (AFR) technology with a movement sensor, which captures images depending on the speed of the capsule’s movement, of the PillCam SB3 (Medtronic, Ltd., Dublin, Ireland; 2–6 images/second) and PillCam Crohn’s capsule (PCC, Medtronic, Ltd.; 4–35 images/second) may increase the diagnostic yield of CD [36,37,38]. Non-white light imaging has been reported to improve the detection rate and visibility of small intestinal lesions by increasing the visualization of surface patterns and color differences in the presence of bile juice and blood [39]. For example, flexible spectral color enhancement (FICE, Fujifilm Corp., Tokyo, Japan) is a digital processing method of white light imaging that emphasizes specific ranges of wavelengths of light in the red, green, and blue spectrum [39]. The FICE wavelength settings were developed with the aims of reducing blue light interference (FICE1), accentuating blood (FICE 2), and strengthening the differences between bile and blood (FICE3) [39]. A contrast capsule (Olympus Corp., Tokyo, Japan), which increases brightness in the blue wavelength range by using a special instrument equipped with a light-emitting diode [39], enables easy detection of areas of bleeding by selecting green and blue data. However, clinical trials of FICE and the contrast capsule so far have yielded controversial results for the detection of gastrointestinal lesions [40,41]. Another optical-digital method similar to contrast capsule, termed narrow-band imaging (Olympus Corp.), which allows better visualization of mucosal surface patterns and superficial capillaries, is being applied to CE in device manufacturing research [42].

### 2.4. Clinical Suspicion of Crohn’s Disease with Negative Conventional Modalities

SBCE is a sensitive tool to detect mucosal abnormalities in the small bowel. Approximately 5–10% of patients have isolated small bowel disease that cannot be detected by conventional ileocolonoscopy [43]. The diagnostic yield of SBCE in patients with SCD has been reported to range from 40% to 70%. For example, SBCE diagnosed CD in 12 (71%) of 17 patients with symptoms such as abdominal pain, anemia, and diarrhea of unknown cause with normal appearance on conventional modalities [44]. SBCE in 20 SCD patients suspected of having small bowel lesions diagnosed CD in 13 patients (65%) [45], showing that CE is effective in diagnosing patients with SCD undetected by conventional diagnostic methods. SBCE detected lesions supporting the diagnosis of CD in 9 (43%) of 21 patients with clinical SCD [46]. This method also increased diagnostic yield by 24% in patients with perianal disease and negative conventional work up, including ileocolonoscopy [47]. These results showed that CE is a useful test for the diagnosis of CD in patients who have not been diagnosed by conventional modalities, such as gastroscopy, ileocolonoscopy, and SBFT.

The presence of biochemical markers in patients with SCD symptoms has been reported to increase the diagnostic yield of SBCE [48]. Although two retrospective studies showed the presence of small bowel inflammation in the majority of ECD patients in remission with biomarkers [49,50], a meta-analysis showed that the likelihood of a positive diagnosis is very low in SCD patients with fecal calprotectin (FC) < 50 μg/g [51]. A recent retrospective study revealed that FC was positively correlated with significant inflammatory activity (Lewis Score, LS ≥ 135; rank correlation = 0.56; *p* < 0.001) [52], further indicating that FC may be a useful marker to select patients with SCD for SBCE. Various FC cut-off values have been reported to be indicative of small bowel CD. For example, 33 (89.2%) of 37 patients with FC ≥ 100 µg/g were found to have an LS ≥ 135 [52]. A recent meta-analysis that included 14 studies suggested that FC ≥ 100 ug/g and LS ≥ 135 cut-off values had diagnostic odds ratios of 8.96 and 10.90, respectively [53].

As normal radiological imaging of the small bowel by SBFT, SBCUS, CTE, and MRE cannot entirely exclude small bowel involvement, SCD patients with normal radiological results but elevated FC and/or unexplained anemia should be considered for additional SBCE [9,13]. Careful monitoring without additional work up may be sufficient for asymptomatic CD patients with negative FC results [54]. Table 1 lists the studies used in this review and their relevant findings. Based on the above results, a diagnostic algorithm using CE in patients with SCD is illustrated in Figure 2.

### 2.5. Clinical Applications for the Assessment of Established Crohn’s Disease

#### 2.5.1. Assessment of Disease Activity, Extent, and Phenotype

CE can be considered for assessment of disease activity, extent, and phenotype in patients with ECD. Classification of CD by disease location showed that 30% of patients had the ileal type, 40% had the ileocolonic type, and 30% had the colonic type [55]. The actual rates of ileal and ileocolonic types may be higher if small bowel disease is evaluated by CE in all patients with ECD. For example, 13/21 (62%) pediatric patients with ECD were found to have more extensive small bowel disease by SBCE [56]. In addition, CE findings led to changes in treatment in 38 (53.5%) of 71 ECD patients [57]. Moreover, CE results have been associated with treatment escalation with thiopurines and/or biologics [58].

Two validated indices, the LS and the Capsule Endoscopy Crohn’s Disease Activity Index (CECDAI), are currently used to assess disease location and activity in the small bowel. Based on assessments of villous edema, ulcers, and stenosis, the LS classifies CD activity from mild to severe [59]. The small bowel is divided into three equal parts based on capsule transit time from the first duodenal image to the first cecal image. For each tertile, a subscore is calculated by multiplying the three index variables, consisting of the number, extent, and descriptors of villous edema and ulcers. The cumulative LS is the sum of the scores for the worst affected tertile and the stenosis score. LS was validated in isolated small bowel CD with strong interobserver agreement [60]. The CECDAI is the sum of the scores of three endoscopic parameters: inflammation (0–5 points), extent of disease (0–3 points), and strictures (0–3 points), for both the proximal and distal segments of the small bowel based on the transit time of the capsule [61]. The CECDAI was validated in a multicenter prospective study of patients with isolated small bowel CD with good interobserver agreement [62]. Cumulative LS and CECDAI strongly correlate with each other and perform similarly in the quantitative assessment of mucosal inflammation in ECD patients. CECDAI < 5.4 and LS < 135 are indicative of mucosal healing, whereas CECDAI > 9.2 and LS ≥ 790 are indicative of moderate-to-severe inflammation [63]. Further studies are needed regarding the usefulness of these SBCE scoring systems in clinical trials and patient practice.

#### 2.5.2. Inflammatory Bowel Disease Unclassified

Approximately 3% of patients with UC have been reclassified as having CD, whereas 0.6–3% of patients initially diagnosed with CD have been reclassified as having UC [14]. In a population-based study, 33% and 17% of patients with indeterminate colitis were reclassified as having UC and CD, respectively [64]. SBCE can be used to classify patients with IBD unclassified (IBDU) and atypical clinical features, although negative SBCE results cannot completely rule out CD. The ability of SBCE to detect small bowel lesions consistent with CD is due to the high sensitivity of SBCE, enabling the reclassification of these lesions [65].

SBCE in IBDU patients has shown varying reclassification rates (16–44%) in several small observational studies. For example, the presence of three or more SB ulcerations was considered diagnostic of CD, resulting in the reclassification of 9 (40.9%) of 22 IBD patients with isolated colitis as having CD [66]. Similar criteria showed that 5 (16.7%) of 30 IBDU patients had lesions suggestive of CD [65], that 19 (15.8%) of 120 patients with IBDU had multiple small bowel ulcerations consistent with CD [67], and that 7 (38.9%) of 18 IBDU patients were diagnosed with CD using SBCE [68]. A retrospective study found that four of five pediatric patients with UC and one of two with indeterminate colitis who underwent CE due to the exacerbation of underlying disease were reclassified as having CD based on newly diagnosed small bowel lesions [56]. A prospective study in 26 pediatric IBDU patients showed that small bowel lesions typical of CD were detected significantly more frequently by CE than by non-endoscopic imaging (43.8% [16/26] vs. 26.9% [7/26]; *p* < 0.05) [69]. In addition, a recent retrospective study showed that SBCE detected significant small bowel lesions (LS ≥ 135) in 9 (25.0%) of 36 patients with IBDU, with all nine confirmed as having CD during follow-up [52].

#### 2.5.3. Treat-to-Target Monitoring in the Management of Crohn’s Disease

As ileocolonoscopic access to the small bowel is difficult, response to treatment should be determined by SBCUS, CTE, and MRE, or by CE in patients with SBCD. Mucosal healing is increasingly recognized as an important treatment goal in patients with IBD [70]. In 2015, the Selecting Therapeutic Targets in Inflammatory Bowel Disease (STRIDE) group suggested a treat-to-target goal of IBD treatment, with the key endpoint being endoscopic mucosal healing [71]. FC has been used to monitor inflammation in the small bowel. The combination of higher FC levels with negative findings on conventional endoscopy suggests the need for further investigations into the presence of active small bowel disease [54]. CE, however, should not be limited to CD patients with positive inflammatory markers because the prediction of significant small bowel inflammation using biomarkers is still poor [72]. In addition, LS of CE and clinical activity as measured by the CDAI did not show significant correlation [73].

Several observational studies evaluated the potential role of SBCE for treat-to-target monitoring in patients with SBCD. SBCE evaluation of the small bowel in 40 CD patients with clinical response showed that clinical response correlated with large ulcers but not with minor mucosal lesions [74]. A prospective study showed that only 8 (15.4%) of 52 patients with clinical remission had small bowel mucosal healing on SBCE [49]. In a large retrospective study, CE findings suggested a change in management for 98 (52.3%) of 187 patients, whereas C-reactive protein and/or FC findings were poorly correlated with significant small bowel inflammation [50]. SBCE detected previously unrecognized disease locations in 40 (51%) of 79 patients in clinical remission, with 27% of patients reclassified as having an advanced phenotype such as B2 or B3, findings that may have had an important impact on both clinical management and long-term prognosis [32]. A meta-analysis that included five observational studies found that mucosal healing detected by SBCE was significantly associated with improved outcomes after 12–24 months, with an OR of 11.06 (95% CI, 3.7–32.7) [75]. Taken together, these findings show that the use of serial CE for longitudinal treat-to-target monitoring of small bowel CD is feasible [76], although future randomized controlled trials are required to confirm the usefulness and reliability of CE for this indication.

#### 2.5.4. Assessment of Post-Operative Recurrence

CE can be used to reliably diagnose post-surgical recurrence. An early small prospective study found that, although CE was less sensitive than ileocolonoscopy in detecting recurrence in the neoterminal ileum, CE was able to detect postoperative lesions outside the scope of ileocolonoscopy in more than two thirds of 32 patients [77]. These findings suggested that SBCE has an advantage in assessing postoperative proximal small bowel recurrence. Another small pilot study revealed that CE can be used to evaluate the postoperative recurrence of small bowel lesions, with LS ≥ 135 in 78% (14/18) of patients 2–3 weeks after surgery for CD [78]. In a meta-analysis of five studies, including 76 CD patients, CE showed a high sensitivity (100%; 95% CI, 91–100%) and accuracy (AUC, 0.94), but a relatively low specificity (69%; 95% CI, 52–83%), for assessing postoperative endoscopic recurrence [79]. Subsequently, CE was found to detect endoscopic recurrence in 11 of 24 patients missed by ileocolonoscopy [80], with residual lesions detected by CE, especially in the distal small intestine, being associated with postoperative clinical recurrence [81]. The lack of high-quality evidence to date, however, has limited the expansion of this indication. A diagnostic algorithm using CE in patients with ECD is illustrated in Figure 3.

#### 2.5.5. Pan-Enteric Capsule Endoscopy for Crohn’s Disease

Pan-enteric capsule endoscopy (PCE) can provide a high diagnostic yield for lesions in the entire gastrointestinal tract in patients with CD [82]. The first-generation CCE (CCE-1), introduced in 2006 (PillCam Colon; Medtronic, Ltd.) [4], was not recommended as a diagnostic tool for IBD, because its sensitivity and NPV were significantly lower than those of colonoscopy [83]. To compensate for these limitations, the second-generation CCE (CCE-2) was introduced in 2009 (PillCam Colon 2; Medtronic, Ltd.) [84]. The CCE-2 has two high-resolution lenses providing a viewing angle of 172° on both sides of the front and back, resulting in a nearly 360° visual field [37]. The CCE-2 also employs an AFR technology that allows it to capture 4–35 images/second based on capsule movement and the FICE 1 mode can be used [37,39]. The PCC has an AFR and lenses on both sides of the capsule, similar to but slightly larger than the CCE-2. The new PCC software version 9 has been shown to be useful in interpreting the results of the PCC system, as well as having a 10% faster read time than previous versions [38]. In addition, this new software includes useful tools that can better estimate lesion severity, extent, and size, providing a more exact evaluation of CD.

Studies have evaluated the diagnostic performances of PCE using CCE-2 or PCC in patients with CD [85]. PCE was successful in CD patients who refused colonoscopy or underwent an incomplete colonoscopy [86]. As expected, PCE had a higher diagnostic yield than ileocolonoscopy (83.3% [55/66] vs. 69.7% [46/66]) for active CD lesions, although three patients were positive by ileocolonoscopy only [87]. When compared with MRE and/or ileocolonoscopy, PCE showed higher sensitivity for active inflammation in the proximal small bowel (97% vs. 71%; *p* = 0.021) but similar sensitivity for inflammation in the terminal ileum and colon [88]. In addition, PCE was feasible in the postoperative surveillance of 50% (6/12) of CD patients with active disease 4–8 months after surgery, whereas ileocolonoscopy detected significant inflammation in 33% (5/15) [89]. PCE was also useful for treat-to-target monitoring in pediatric CD patients, with changes in treatment based on PCE findings [90]. Inflammation in patients with ileocolonic CD can be assessed by CECDAIic score, which was based on CECDAI score [91], and inflammation in patients with pan-enteric CD can also be evaluated by determining PCC score, which was based on LS [92]. Both of these scores showed high interobserver reliability and strong correlation with FC levels [92,93]. A recent European multicenter observational study found that complete examination by PCE in 79 (85%) of 93 patients resulted in changes in therapy in 36 (38.7%) patients [94]. The sensitivity and specificity of PCE for small bowel lesions (84.2–90.0% and 63.4–87.5%, respectively) and large bowel lesions (80.0–90.0% and 72.2–84.7%, respectively) were comparable to the results of double balloon enteroscopy in patients with CD [95]. Acceptable bowel preparation rates for PCE were reported to range from 63.9% to 100% [85]. At present, the applications and values of PCE for CD remain unclear yet due to lack of large scale and/or randomized controlled trials.

## 3. Ulcerative Colitis

CCE can also be used to evaluate patients with UC [17,96,97,98,99,100]. According to STRIDE, the treatment goals in UC include a Mayo Endoscopic Score (MES) of 0 as the optimal target and a MES of 1 as the minimal target [101]. However, performing a colonoscopy every year or every other year may be difficult in clinical practice due to technical difficulties or the risk of flare-up after colonoscopy. CCE-2 was used to evaluate the severity of mucosal inflammation in 29 patients with remission to moderate UC [17]. CCE-2 findings were recorded for 8 h, starting from capsule ingestion with subsequent bowel preparation using 2 L low-volume polyethylene glycol (PEG) with prokinetics. The total colon was observed in 20 (69.0%) patients, and the extent of disease could be determined. Although bowel preparation was insufficient (fair or poor) in >50% of these patients, Matts endoscopic score showed a stronger correlation when compared with colonoscopy (*p* = 0.797) [17]. These findings showed that CCE-2 could be used to evaluate patients with UC, although improvements in bowel preparation were required. A modified bowel preparation procedure, consisting of a maximum of 2.2 L lavage solution of PEG and magnesium citrate taken in two or three divided doses, resulted in a total colon observation rate of 85.0% (17/20 of patients), which was considered to be insufficient [96]. A low-volume PEG solution containing ascorbic acid (PEG-ASC) may reduce the volume of preparation for UC. For example, a maximum dose of 3.0 L liquid (2 L PEG-ASC and 1 L free water with a small amount of castor oil) increased the total colon observation rate to 93.9% (31/33) [99]. CCE-2 and colonoscopy showed good correlations in measuring MES (intraclass correlation coefficient (ICC), 0.69; 95% CI, 0.46–0.81; *p* < 0.001) and UC Endoscopic Index Score (UCEIS) (ICC, 0.64; 95% CI, 0.38–0.78; *p* < 0.001) in 108 patients with UC [97]. Another recent study also showed a moderate agreement for the disease extent of UC and a very good overall agreement between these two diagnostic modalities for disease activity of UC [100].

A scoring system for CCE, consisting of a modification of the UCEIS, may be used to assess the severity of capsule scoring of UC (CSUC) [102]. The CSUC can be determined as the sum of the vascular pattern, bleeding, and erosions and ulcers, with scores ranging from 0 to 14 points. The correlation coefficients of CSUC with biomarkers and clinical score were similar to those of the UCEIS. A prospective study in 41 UC patients in clinical remission who underwent CCE-2 showed that the CSUC was significantly higher in the 12 patients who relapsed within 1 year than in the 29 patients who remained in clinical remission (2.83 ± 1.95 vs. 0.72 ± 1.00; *p* < 0.01) [103]. Moreover, CSUC ≥ 1 was found to be prognostic of relapse, with a sensitivity of 83.3%, a specificity of 58.6%, and an accuracy of 0.82, and to be superior to the prognostic ability of fecal biomarkers [103]. However, the CSUC requires further validation prior to its widespread use to accurately determine the disease activity and extent of UC.

There are several barriers to applying CCE to UC in clinical practice. In general, patients consider CCE to be more tolerable and preferable than ileocolonoscopy [97]. However, a recent meta-analysis of patient-reported outcomes and preference showed no differences in preference and tolerability between CCE and colonoscopy [104]. As CCE is performed when the colon is deflated, the severity of lesions may be overestimated or underestimated when compared with colonoscopy [37]. In addition, CCE is not applicable diagnostically for surveillance of dysplasia-associated lesion or masse in UC [37]. Studies included in this review along with their pertinent findings are summarized in Table 2, and typical CCE findings in patients with UC are shown in Figure 1B.

## 4. Conclusions

The development of CE over the past 20 years has contributed greatly to the understanding of the pathophysiology of small bowel diseases, including IBD. CE is undergoing development to improve its diagnostic yield and accuracy. These include developments designed to increase frame rate and image resolution, widen the viewing angle, increase image resolution, and extend battery life. CE is not a diagnostic reference paradigm to confirm small bowel CD but rather an evaluation tool to detect clues necessary for diagnosing CD in patients with SCD. CE may also be a useful diagnostic modality to determine the causes of non-response to treatment in patients with ECD, as well as to monitor treat-to-target strategies and detect postoperative recurrences. Due to its non-invasiveness and convenience, the scope of CE applications is expanding to patients with UC and pan-enteric CD instead of ileocolonoscopy. Artificial intelligence (AI) is already starting to affect the reading protocol of CE and will provide solutions to tedious reading problems. The coupling of AI and CE may provide more accurate information on patients with anomalies in a single frame that are easy to miss, as well as new gastrointestinal mapping information to localize the capsule. The development of AI may expand the utilization of CE in IBD over the next two decades.

## Figures and Tables

**Figure 1 diagnostics-11-02240-f001:**
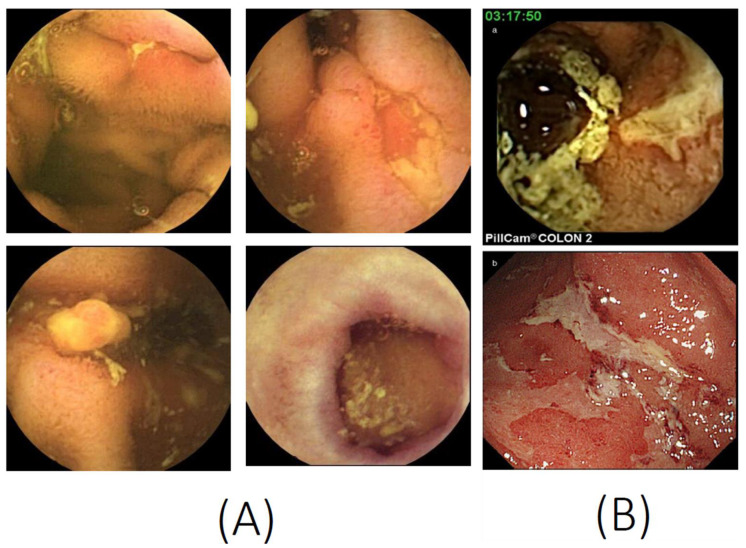
Capsule endoscopy images. (**A**) Small bowel capsule endoscopy features of Crohn’s disease, such as ulcers, longitudinal ulcers, inflammatory polyps, and scars. (**B**) A colon capsule endoscopy image (**a**) consistent with the conventional colonoscopy image (**b**) of ulcerative colitis (photocopies from Hosoe N, et al. J. Gastroenterol. Hepatol. 2013, 28, 1174–1179, with permission from John Wiley and Sons [17]).

**Figure 2 diagnostics-11-02240-f002:**
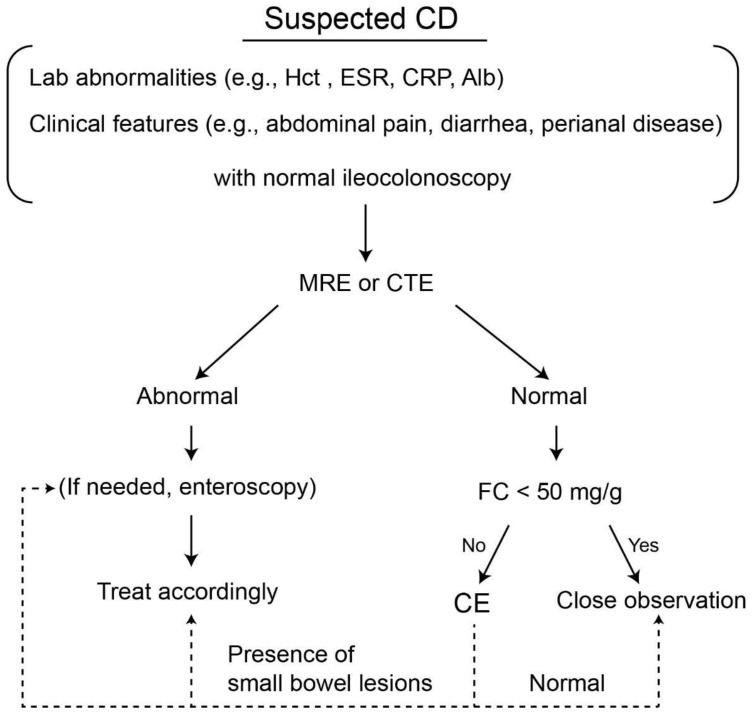
Suggested diagnostic algorithm for the use of small bowel capsule endoscopy in patients with suspected Crohn’s disease. CD, Crohn’s disease; Hct, hematocrit; CRP, C-reactive protein; Alb, albumin; MRE, MR enterography; CTE, CT enterography; FC, fecal calprotectin, CE, capsule endoscopy.

**Figure 3 diagnostics-11-02240-f003:**
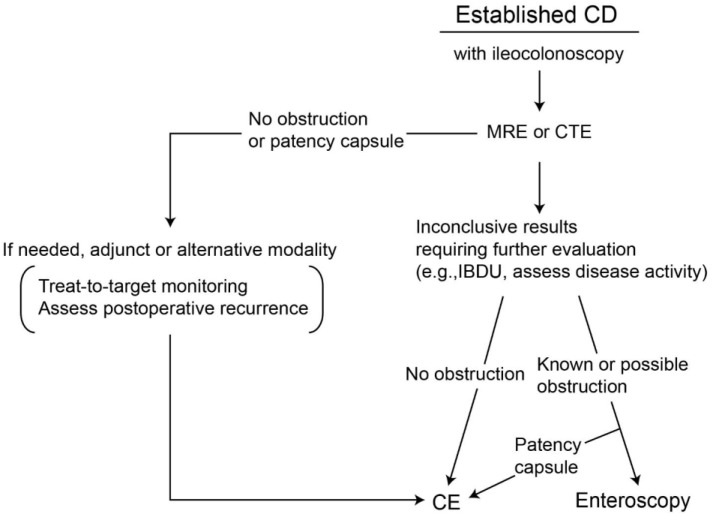
Suggested diagnostic algorithm for the use of small bowel capsule endoscopy in patients with established Crohn’s disease. CD, Crohn’s disease; MRE, MR enterography; CTE, CT enterography; IBDU, unspecified inflammatory bowel disease; CE, capsule endoscopy.

**Table 1 diagnostics-11-02240-t001:** Summary of results of studies on the clinical usefulness of capsule endoscopy for the evaluation of suspected Crohn’s disease.

Author Year	Study Design	Topic	Study Papulation	Findings
Fireman 2003 [44]	Prospective observational	Effectiveness of CE in SCD	17 SCD undetected by conventional modalities	12 (71%) patients were diagnosed with SBCD
Herrerias 2003 [46]	Prospective observational	Effectiveness of CE in SCD	21 SCD undetected by conventional modalities	Pathologic findings in 12 (57%) patients 9 (43%) patients were diagnosed with SBCD
Ge 2004 [45]	Prospective observational	Effectiveness of CE in SCD	20 SCD undetected by conventional modalities	13 (65%) patients were diagnosed with SBCD
De Bona 2006 [48]	Prospective observational	Effectiveness of CE in SCD with increased biochemical markers	Group 1: 12 clinical SCD only Group 2: 26 clinical SCD + positive biochemical markers	Overall diagnostic yield: 39% (13 diagnostic and 2 suspicious for SBCD) Group 1 vs. Group 2: 8.3% vs. 46.2% (*p* = 0.022)
Adler 2012 [47]	Prospective observational	Role of CE in patients with persistent perianal disease	25 patients with perianal disease and negative conventional modalities	6 patients (24%) were diagnosed with SBCD
Kopylov 2016 [51]	Meta-analysis	Predictive value of FC to diagnose SBCD by CE findings in SCD	5 studies (305 patients) with SCD and negative ileocolonocsopy	FC ≥ 50 ug/g: sensitivity 89%, specificity 55%, DOR 10.3, NPV 92% in predicting CE findings for SBCD FC < 50 ug/g: very low PLR
Bar-Gil Shitrit 2017 [54]	Prospective observational	Predictive value of FC to diagnose SBCD by CE findings	68 patients underwent CE for any indications with negative ileocolonoscopy	Median FC: SBCD (23 patients) vs. non-SBCD (45 patients) = 169 mg/kg vs. 40 mg/kg (*p* = 0.004)
Monteiro 2018 [52]	Retrospective observational	Predictive value of FC to diagnose SBCD by CE findings in SCD	75 SCD with negative ileocolonoscopy	In 37 patients with FC ≥ 100 µg/g, an LS ≥ 135 was found in 33 (89.2%)
Jung 2021 [53]	Meta-analysis	Predictive value of FC to diagnose SBCD by CE findings in SCD	8 studies (696 patients) with SCD and negative ileocolonoscopy	FC ≥ 100 ug/g: sensitivity 75%, specificity 74%, DOR 9.0 in predicting CE findings for SBCD

CE, capsule endoscopy; SBCD, small bowl Crohn’s disease; SCD, suspected Crohn’s disease; FC, fecal calprotectin; DOR, diagnostic odds ratio; NPV, negative predictive value; PLR, positive likelihood ratio; LS, Lewis Scores.

**Table 2 diagnostics-11-02240-t002:** Summary of results of studies on the clinical applications of colon capsule endoscopy for ulcerative colitis.

Author Year	Study Design	Topic	Study Papulation	Bowel Cleansing	Concomitant Prokinetics	Major Findings
Hosoe 2013 [17]	Prospective observational	Assessment disease activity of UC using CCE-2	29 UC patients	2 L PEG	Mosapride, metoclopramide	Procedure completion in 20 (69%) patients Acceptable cleansing < 50% Correlation with Matts endoscopic scores (*p* = 0.80 ^†^)
Usui 2014 [96]	Prospective observational	Bowel cleansing regimen	20 UC patients	2.2 L PEG + magnesium citrate	Mosapride, metoclopramide	Procedure completion in 17 (85.0%) patients Acceptable cleansing: 60%
Shi 2017 [97]	Prospective observational	Assessment disease activity of UC using CCE-2	150 UC patients	4 L PEG + phosphate soda	Metoclopramide	Procedure completion in 109 (73%) patients Acceptable cleansing: 66% Correlation with MES (ICC, 0.69; 95% CI, 0.46–0.81; *p* < 0.001) Correlation with UCEIS (ICC 0.64; 95% CI, 0.38–0.78; *p* < 0.001)
Takano 2018 [98]	Prospective observational	Assessment disease activity of UC using CCE-2	30 UC patients	2 L PEG + 1.4 L water + magnesium citrate	Mosapride	Procedure completion in 17 (85.0%) patients Acceptable cleansing: 73%
Okabayashi 2018 [99]	Prospective observational	Factors associated with colonic transit time and acceptability of CCE-2	33 UC patients	2 L PEG-Asc + 1 L water + 20 mL castor oil	Metoclopramide	Procedure completion in 31 (94%) patients Acceptable cleansing: 77% Median colonic transit time: 119 min Factors associated with colonic transit time: high MES and/or UCEIS
Adler 2019 [100]	Prospective observational	Assessment disease activity of UC using CCE-2	23 UC patients	2 L PEG + 0.75 L trisulfate + 1.5 L water	Metoclopramide	Percent agreement for disease extent (57%; *p* = 0.43 ^‡^) Percent agreement for MES (96%; *p* = 0.86 ^‡^)

^†^ Spearman’s rank correlation coefficient; ^‡^ Kappa coefficient CCE, colon capsule endoscopy; UC, ulcerative colitis; PEG, polyethylene glycol; PEG-Asc, PEG solution containing ascorbic acid; MES, Mayo Endoscopic Score; UCEIS, Ulcerative Colitis Endoscopic Index Score; ICC, intraclass correlation coefficient.

## Data Availability

Not applicable.

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
