# Peer review of "Capsule Endoscopy in Inflammatory Bowel Disease: When? To Whom?"

_diagnostics, 2021, doi:10.3390/diagnostics11122240_

Round 1

Reviewer 1 Report

Thanks for submitting this informative, well-written review. I only have some minor comments on this. 

  1. Introduction - Please add sentences on the increasing disease burden of IBD in the World or in Asia including your countries. This would be helpful for our readers to understand the need for CE in the field of IBD.
  2. You use both SBCUS and SBUS for small bowel ultrasound. Please use the uniform abbreviation throughout the manuscript.
  3. In page 4, lines 139-140, same sentences were repeated. Please delete one of them. 
  4. In page 5, lines 177, "..algorithm using CCE...' should be revised to "...algorithm using SBCE..". 
  5. In page 7, line 214, I think 'inflammatory bowel disease unclassified' would be better than 'unclassified inflammatory bowel disease".
  6. In page 7, lines 242-244, you can replace this reference with the recent update of STRIDE II (Gastroenterology 2021;160(5):1570-1583).
  7. In page 8, lines 268, I think the title would be revised to "4) Assessment of post-operative recurrence". 

Author Response

Thank you for your kind review and valuable comments.

Thanks for submitting this informative, well-written review. I only have some minor comments on this. 

  1. Introduction - Please add sentences on the increasing disease burden of IBD in the World or in Asia including your countries. This would be helpful for our readers to understand the need for CE in the field of IBD.

Response: Thank you for your valuable comments. We added content related to your comment to the introduction and marked it in red color as follows;

The incidence of inflammatory bowel disease (IBD) is relatively high but rather stable in Western countries; however, the incidence and prevalence of IBD are rapidly increasing in Asia, Eastern Europe, and South America, and this shift in the epidemiology of IBD indicates that it has become a global disease, whose increasing incidence is burdening health services in recent years [5-8]. Therefore, the use of CE for the diagnosis and management of IBD is becoming more frequent and its implementation is considered a priority in the field of IBD. This review will describe the clinical applications and value of CE in the diagnosis and management of inflammatory bowel disease.

  1. You use both SBCUS and SBUS for small bowel ultrasound. Please use the uniform abbreviation throughout the manuscript.

Response: Thank you for your attentive review and comments. We unified the abbreviation as SBCUS. 

  1. In page 4, lines 139-140, same sentences were repeated. Please delete one of them. 

Response: Thank you for your attentive review and comments. We deleted one of the duplicate sentences.

  1. In page 5, lines 177, "..algorithm using CCE...' should be revised to "...algorithm using SBCE..". 

Response: Thank you for your attentive review and comments. We changed CCE to SBCE as your comment. 

  1. In page 7, line 214, I think 'inflammatory bowel disease unclassified' would be better than 'unclassified inflammatory bowel disease".

Response: Thank you for your valuable comments. We changed 'inflammatory bowel disease unclassified' to 'unclassified inflammatory bowel disease as your comment. 

  1. In page 7, lines 242-244, you can replace this reference with the recent update of STRIDE II (Gastroenterology 2021;160(5):1570-1583).

Response: Thank you for your attentive review and comments. We changed reference as your comment.

  1. In page 8, lines 268, I think the title would be revised to "4) Assessment of post-operative recurrence". 

Response: Thank you for your attentive review and comments. We changed the title as your comment.

Reviewer 2 Report

As a biomedical engineer, I would expect an article like a review of tips on how to improve current capsule endoscopes to increase their diagnostic capabilities. However, the article presents only a dry comparison of the effectiveness of endoscopic capsules to other methods, without indicating which additional sensors should be equipped with the capsule in the future. Quite generally, it has been found that higher image resolution as well as frame rate are desirable.

On lines 337 and 340 there is a repeated sentence

Author Response

Thank you for your kind review and valuable comments.

  1. As a biomedical engineer, I would expect an article like a review of tips on how to improve current capsule endoscopes to increase their diagnostic capabilities. However, the article presents only a dry comparison of the effectiveness of endoscopic capsules to other methods, without indicating which additional sensors should be equipped with the capsule in the future. Quite generally, it has been found that higher image resolution as well as frame rate are desirable.

Response: Thank you for your valuable comments.

1) We created a separate paragraph in the text to add related contents as your comment and marked it in red color as follows;

Increasing diagnostic capability of capsule endoscopy

CE is being developed in several ways, including the development of CE instruments with higher frame rates and increased image resolution, which should increase the possibility of obtaining higher diagnostic yield and accuracy than in the past [34]. For example, the adaptive frame rate (AFR) technology with a movement sensor, which captures images depending on the speed of the capsule's movement, of the PillCam SB3 (Medtronic, Ltd., Dublin, Ireland; 2-6 images/second) and PillCam Crohn’s capsule (PCC, Medtronic, Ltd.; 4-35 images/second) may increase the diagnostic yield of CD [35-37]. Non-white light imaging has been reported to improve the detection rate and visibility of small intestinal lesions by increasing the visualization of surface patterns and color differences in the presence of bile juice and blood [38]. For example, flexible spectral color enhancement (FICE, Fujifilm Corp., Tokyo, Japan) is a digital processing method of white light imaging that emphasizes specific ranges of wavelengths of light in the red, green, and blue spectrum [38]. The FICE wavelength settings were developed with the aims of reducing blue light interference (FICE1), accentuating blood (FICE 2), and strengthening the differences between bile and blood (FICE3) [38]. A contrast capsule (Olympus Corp., Tokyo, Japan), which increases brightness in the blue wavelength range by using a special instrument equipped with a light-emitting diode [38], enables easy detection of areas of bleeding by selecting green and blue data. However, clinical trials of FICE and the contrast capsule so far have yielded controversial results for the detection of gastrointestinal lesions [39,40]. Another optical-digital method similar to contrast capsule, termed narrow-band imaging (Olympus Corp.), which allows better visualization of mucosal surface patterns and superficial capillaries, is being applied to CE in device manufacturing research [41].

2) We described the related contents as your comments in the sub-topic section of 'Pan-enteric capsule endoscopy for Crohn's disease' and marked it in red color as follows;

The first-generation CCE (CCE-1), introduced in 2006 (PillCam Colon; Medtronic, Ltd.) [4], was not recommended as a diagnostic tool for IBD, because its sensitivity and NPV were significantly lower than those of colonoscopy [82]. To compensate for these limitations, the second-generation CCE (CCE-2) was introduced in 2009 (PillCam Colon 2; Medtronic, Ltd.) [83]. The CCE-2 has two high-resolution lenses providing a viewing angle of 172° on both sides of the front and back, resulting in a nearly 360° visual field [36]. The CCE-2 also employs an AFR technology that allows it to capture 4–35 images/second based on capsule movement and the FICE 1 mode can be used [36,38]. The PCC has an AFR and lenses on both sides of the capsule, similar to but slightly larger than the CCE-2. The new PCC software version 9 has been shown to be useful in interpreting the results of the PCC system, as well as having a 10% faster read time than previous versions [37]. In addition, this new software includes useful tools that can better estimate lesion severity, extent, and size, providing a more exact evaluation of CD.

3) We added contents related to your comments in the ‘Conclusions’ and marked it in red color as follows;

CE is undergoing development to improve its diagnostic yield and accuracy. These include developments designed to increase frame rate and image resolution, widen the viewing angle, increase image resolution, and extend battery life

  1. On lines 337 and 340 there is a repeated sentence.

Response: Thank you for your attentive review and comments. We deleted one of the duplicate sentences.